# Dichotomy of the BSL phosphatase signaling spatially regulates MAPK components in stomatal fate determination

Xiaoyu Guo [1], Xue Ding[1] & Juan Dong [1,2]✉

MAPK signaling modules play crucial roles in regulating numerous biological processes in all eukaryotic cells. How MAPK signaling specificity and strength are tightly controlled remains a major challenging question. In *Arabidopsis* stomatal development, the MAPKK Kinase YODA (YDA) functions at the cell periphery to inhibit stomatal production by activating MAPK 3 and 6 (MPK3/6) that directly phosphorylate stomatal fate-determining transcription factors for degradation in the nucleus. Recently, we demonstrated that BSL1, one of the four BSL protein phosphatases, localizes to the cell cortex to activate YDA, elevating MPK3/6 activity to suppress stomatal formation. Here, we showed that at the plasma membrane, all four members of BSL proteins contribute to the YDA activation. However, in the nucleus, specific BSL members (BSL2, BSL3, and BSU1) directly deactivate MPK6 to counteract the linear MAPK pathway, thereby promoting stomatal formation. Thus, the pivotal MAPK signaling in stomatal fate determination is spatially modulated by a signaling dichotomy of the BSL protein phosphatases in *Arabidopsis*, providing a prominent example of how MAPK activities are integrated and specified by signaling compartmentalization at the subcellular level.

[1] Waksman Institute of Microbiology, Rutgers, The State University of New Jersey, Piscataway, NJ 08854, USA. [2] Department of Plant Biology, Rutgers, The State University of New Jersey, New Brunswick, NJ 08901, USA. ✉email: dong@waksman.rutgers.edu

Mitogen-Activated Protein Kinases (MAPKs) are universally important signaling molecules that transduce extracellular signals to induce cellular responses. In plants, a MAPK cascade usually is composed of three tiers, including MAPK Kinase Kinase (MAPKKK), MAPK Kinase (MKK), and MAPK (MPK), that relay, amplify and integrate signals from a diverse range of stimuli to elicit appropriate cellular responses and control developmental processes[1], such as stomatal development and patterning. Stomata are microscopic pores in the epidermis of aerial organs of a plant. Stomatal movement allows gas exchange between a plant and the environment and restricts excessive water evaporation from the plant. As the number of stomata needs to be optimal for different plant species growing in the changing environment, stomatal development is heavily shaped by environmental cues[2,3].

In the model plant *Arabidopsis*, mutating components in the key canonical MAPK cascade, including the *MAPKKK YODA* (*YDA*), *MAPKK 4 and 5* (*MKK4/5*), or *MAPK 3 and 6* (*MPK3/6*), results in comparable phenotypes of overly proliferating and abnormally patterned stomatal guard cells[4,5]. Previous studies suggest that the MAPKKK YDA is activated by the EPIDERMAL PATTERNING FACTOR (EPF) peptides via their interaction with the ERECTA receptor-like kinase (RLK) family (ERf)[6] that partner with the receptor-like protein TOO MANY MOUTHS (TMM) and the RLK SOMATIC EMBRYOGENESIS RECEPTOR KINASES (SERKs)[7,8]. Activated MPK3/6 directly phosphorylate the master regulators of stomatal development, the basic helix–loop–helix (bHLH) transcription factor SPEECHLESS (SPCH) and its partners ICE1/SCREAM(SCRM) and SCRM2, leading to their degradation[9–11]. Activities of YDA and MKK4/5 can be negatively regulated by the GSK3-like BIN2 kinase in stomatal development[12,13]. However, we know little about how different tiers of the MAPK cascade are precisely regulated and integrated for signaling strength and specificity in plant developmental processes and environmental responses.

The BSU1 (*bri1* Suppressor 1) phosphatase was initially characterized to promote Brassinosteroid (BR) responses by directly dephosphorylating and deactivating the BIN2 kinase, the key negative regulator of BR signaling in plants[14]. We recently showed that members of the BSU1-like (BSL) family of Ser/Thr protein phosphatases play key roles in stomatal development[15]. Three BSL proteins (BSL1, BSL2, and BSL3) directly interact with the polarity protein BREAKING OF ASYMMETRY IN THE STOMATAL LINEAGE (BASL) to enable the progression of stomatal asymmetric cell division[15]. BSL1 becomes polarized at the cortex of meristemoid mother cells (MMCs) and acts at the plasma membrane (PM) to activate YDA kinase activity and promote stomatal asymmetric cell-fate differentiation[15].

In this study, we uncover that the BSL signaling conveys novel dichotomy effects, through which stomatal production in *Arabidopsis* can be positively and negatively controlled. We demonstrate that at the plasma membrane (PM), all four BSL members contribute to the positive regulation of the MAPKKK YDA, whereas BSL2, BSL3, and BSU1 enforce a negative regulation on MPK6 in the nucleus. Thus, we establish a signaling dichotomy of the BSL proteins occurring at distinct subcellar locations regulates a linear MAPK cascade, providing a prominent example of spatial control of MAPK signaling in decision-making during plant development.

## Results

### Positive and negative regulation of BSL members in stomatal development.
In *Arabidopsis*, the stomatal lineage is initiated from the conversion of a protodermal cell into a meristemoid mother cell (MMC), which divides asymmetrically to generate a

meristemoid (M, small) and a stomatal lineage ground cell (large, SLGC) (Fig. 1a). Both M and SLGC have the potential to further divide to eventually differentiate into stomatal guard cells and non-stomatal pavement cells in the epidermis, respectively. It has been previously shown that mutating all four *BSL* genes (*bsl-q*, containing T-DNA insertional *bsu1* and *bsl1* combined with knocking-down *amiRNA-BSL2;3*) results in overproliferated stomatal lineage and guard cells[12,15]. The phenotype of *bsl-q* suggested the collective role of all four *BSL* genes in the inhibition of stomatal production. While none of the single *bsl* mutants (T-DNA insertional lines) showed discernable stomatal and growth defects, our previous work also demonstrated that mutating more than two of the three BASL-interacting members, i.e., BSL1, BSL2, and BSL3, led to slightly increased stomatal production, while mutating all three led to moderately increased stomatal production[15] (Fig. 1b). To thoroughly determine the contribution of BSU1 in the BSL1/BSL2/BSL3-mediated negative regulation, we compared *bsl-q* with the null triple mutant *bsl1;bsl2;bsl3* and found that the quadruple mutant indeed showed increased stomatal production (quantified by stomatal lineage index, SLI, measured in the central region of adaxial side cotyledons in 5-day seedlings) (Fig. 1b). Moreover, the quadruple mutant was featured with severely clustered stomata often appearing as patches at the leaf edges (Supplementary Fig. 1a) that were not observed in the triple mutants. As the *bsl-q* mutant expresses *amiR-BSL2;3* that silences the two close homologs (*BSL2* and *BSL3*), the possibility of off-targeting was ruled out by the detection of wild-type level of *BSL1* and *BSU1* transcripts in plants containing *amiR-BSL2;3* and by comparable phenotypes of *bsl1;bsl2;bsl3* and *bsl1;amiRNA-BSL2;3* (Supplementary Fig. 1b, c). Thus, our data indicated that BSU1 positively contributes to the BSL1/BSL2/BSL3-mediated inhibition of stomatal production in *Arabidopsis* (Supplementary Fig. 1).

However, while we characterized all possible combinations of *bsl* double and triple mutations, surprisingly we found that differential combination of *bsl* mutations resulted in distinct phenotypes with regards to stomatal production (Fig. 1b, Supplementary Fig. 2). While *bsl1;bsl2;bsl3* produced more stomatal lineage cells[15] (Fig. 1b), mutating the other three members, i.e., *bsl2;bsl3;bsu1* led to an opposite phenotype— lowered stomatal production (Fig. 1b, quantification based on stomatal lineage index). Combination of two mutations among these three genes (*BSL2, BSL3, BSU1*) also generated similarly lowered stomatal production in the mutants, e.g., *bsl2;bsu1* and *bsl3;bsu1* (Fig. 1b, Supplementary Fig. 2f, g). Thus, the results suggest that the combined function of *BSL2, BSL3,* and *BSU1* confers an opposite regulation to promote stomatal production.

The strikingly opposite phenotypes of the *bsl-q* mutant (high production) *vs.* the triple *bsl2;bsl3;bsu1* (low production) suggested that *BSL1* may play a prominent role in suppressing stomatal production. Indeed, while *bsl2;bsl3* was almost wild-type looking, the addition of *bsl1* generated stomatal overproduction in *bsl1;bsl2;bsl3* (Supplementary Fig. 2e), whereas the addition of *bsl1* reversed the phenotype of low stomatal production in *bsl2;bsu1* and *bsl3;bsu1* to the wild-type levels in *bsl1;bsl2;bsu1* and *bsl1;bsl3;bsu1* mutants, respectively (Supplementary Fig. 2f, g). Thus, results of genetic analyses suggested that among the four *BSL* members, *BSL1* appeared to be the major regulator in inhibition of stomatal production.

Also, among *BSL2, BSL3,* and *BSU1* identified as positive regulators, *BSU1* was revealed as the major player in promoting stomatal production. Our data showed that, while *bsl2;bsl3* is wild-type looking, either *bsl2;bsu1* or *bsl3;bsu1* exhibited lowered stomatal production (Supplementary Fig. 2e-g). Furthermore, in *bsl1;bsl3;bsu1*, the addition of *bsu1* reversed the stomatal overproduction phenotype of *bsl1;bsl3* to the wild-type level (Fig. 1b,

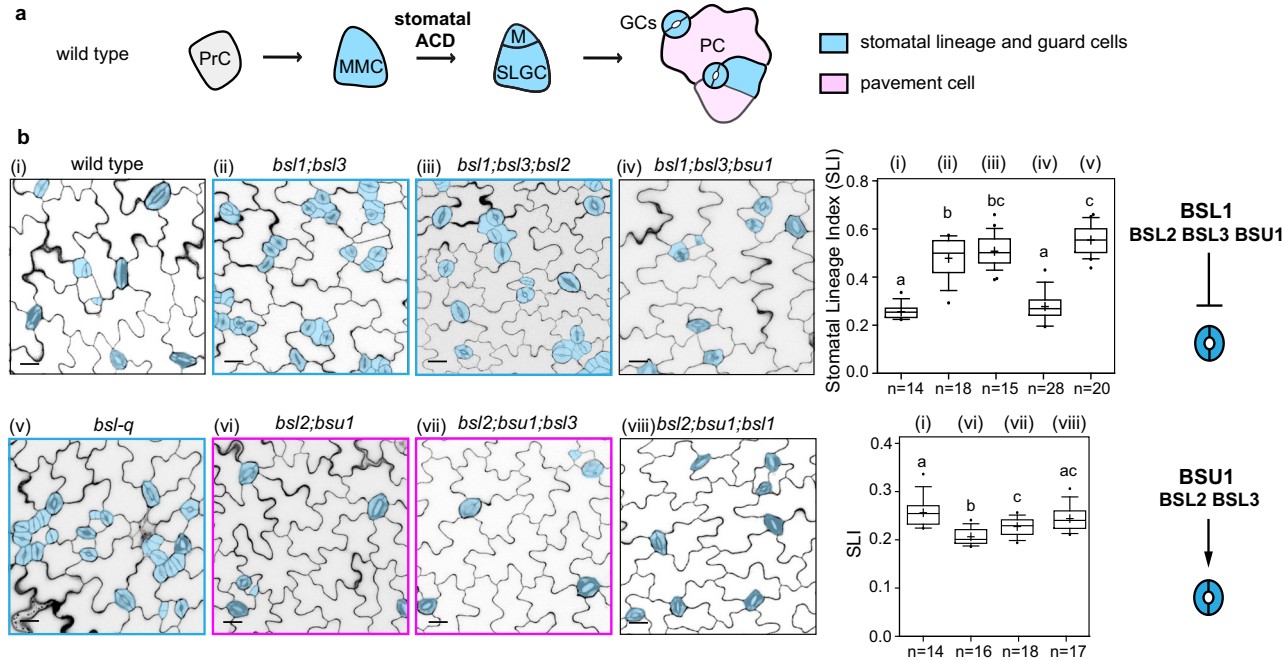

**Fig. 1 Opposing regulation of BSL genes in stomatal development. a** Schematic diagrams to describe stomatal development in *Arabidopsis*. PrC (Protodermal Cell), MMC (Meristemoid Mother Cell), M (Meristemoid), SLGC (Stomatal Lineage Ground Cell), GCs (Guard Cells), and PC (Pavement Cell). Blue shade, stomatal lineage, and guard cells; Pink shade, non-stomatal pavement cells. **b** Left: (i–viii) Confocal images show stomatal phenotypes in adaxial cotyledon epidermis of a 5-dpg (day post germination) wild-type or a *bsl* mutant plant. Stomatal lineage and guard cells are manually traced (blue). Scale bars, 20 μm. Blue boxes highlight the genotypes producing more stomatal lineage cells and pink boxes highlight the genotypes producing fewer stomatal lineage cells. Middle: Plot boxes show Stomatal Lineage Index (# stomata lineages relative to # total cells) of the designated genotypes. Box plot shows first and third quartiles, median (line), and mean (cross). Letters were assigned by ANOVA and Tukey's multiple comparison test ($P \leq 0.01$). Exact $P$ values obtained by unpaired *t*-test are $7.58^{e-10}$ for i *vs.* ii, $7.569^{e-20}$ for i *vs.* iii, 0.2435 for i *vs.* iv, $5.165^{e-18}$ for i *vs.* v, $1.77718^{e-05}$ for i *vs.* vi, 0.0045948 for i *vs.* vii, 0.2213566 for i *vs.* viii, 0.2179877 for ii *vs.* iii, $9.5308^{e-09}$ for ii *vs.* iv, 0.003676 for ii *vs.* v, 0.01535 for iii *vs.* v, 0.0014766 for *vi vs.* vii, $1.33323^{e-05}$ for vi *vs.* viii, 0.0336666 for vii *vs.* viii. *n*, number of cotyledons used for quantification of each genetic background. Right: Graphics describe differential regulation of the *BSL* genes in stomatal development. A complete panel of *bsl* mutant analysis can be found in Supplementary Fig. 2.

Supplementary Fig. 2c). Thus, our genetic data suggested that, among the three positive regulators (*BSL2*, *BSL3*, and *BSU1*), *BSU1* appears to play a predominant role in promoting stomatal production.

**Differential overexpression phenotypes and subcellular localization of the BSL proteins.** Based on our mutant analyses described above, we proposed that *BSL2*, *BSL3*, and *BSU1* participate in both negative and positive regulations of stomatal production, while *BSL1* only participates in the negative regulation pathway. To further test this hypothesis, we overexpressed each of the individual *BSL* members driven by the stomatal lineage-specific *TMM* promoter[7] in *Arabidopsis* wild-type plants (*TMMp*::BSL1 was first reported in ref. [15]). Results showed that overexpression of *BSL1* strongly suppressed stomatal formation, whereas overexpression of *BSL2* and *BSL3* induced moderately increased stomatal production, and overexpression of *BSU1* induced severely proliferative stomatal lineage cells (Fig. 2a, Supplementary Fig. 3a, transgene expression levels examined in Supplementary Fig. 3b). Thus, consistent with the loss-of-function mutant analysis, the overexpression data suggested that BSL1 is a strong inhibitor of stomatal production, whereas all the other three BSL members can promote stomatal production, with BSU1 demonstrating the highest activity.

Next, we analyzed possible differences among the four BSL proteins that might explain their differentially functional contribution. Based on the protein-sequence alignment, we did not observe striking domain features that may make BSL1 distinct, except that BSL1 contains a unique "GTLDE" motif required for its membrane association (Supplementary Fig. 3c)[16]. Indeed, we

observed that the expression of BSL1-YFP driven by the native promoter showed predominantly cytoplasmic and plasma-membrane localization in the leaf epidermal cell. As reported previously, BSL1-YFP is also polarized in stomatal lineage cells and is absent from the nucleus (Fig. 2b)[15]. In contrast, all the other three BSL members (BSL2, BSL3, and BSU1) showed both cytoplasmic and nuclear localization[15] (Fig. 2b, blue arrowheads show nuclear localization), with BSU1-YFP more abundantly partitioned in the nucleus (Fig. 2c, quantified for nuclear/membrane partition). Correspondingly, overexpression of *BSL2/ 3* and *BSU1* generated differential levels of stomatal production (*BSL2/3* moderate and *BSU1* severe) (Supplementary Fig. 3a). Thus, we hypothesized that the PM/cytoplasmic pool of all four BSL proteins contributes to the negative regulation of stomatal production, whereas the nuclear partition of BSL2, BSL3, and BSU1 contributes to the positive regulation.

**BSL phosphatases promote stomatal production in the nucleus.** To explicitly assay the functional contribution of the BSL proteins at the subcellular level, we first used BSL2, a family member functioning at both places, as an example to examine its nuclear/ membrane partition in developing stomatal lineage cells. By co-expression with the BASL polarity protein[15], we found that BSL2-mRFP is more enriched in the nucleus when BASL is not polarized, but BSL2-mRFP became more membrane-associated when BASL is polarized (Supplementary Fig. 4). The results indicated the subcellular distribution thus the function of BSL2 can be dynamically and spatially regulated by developmental regulators. Next, we engineered individual proteins, either by

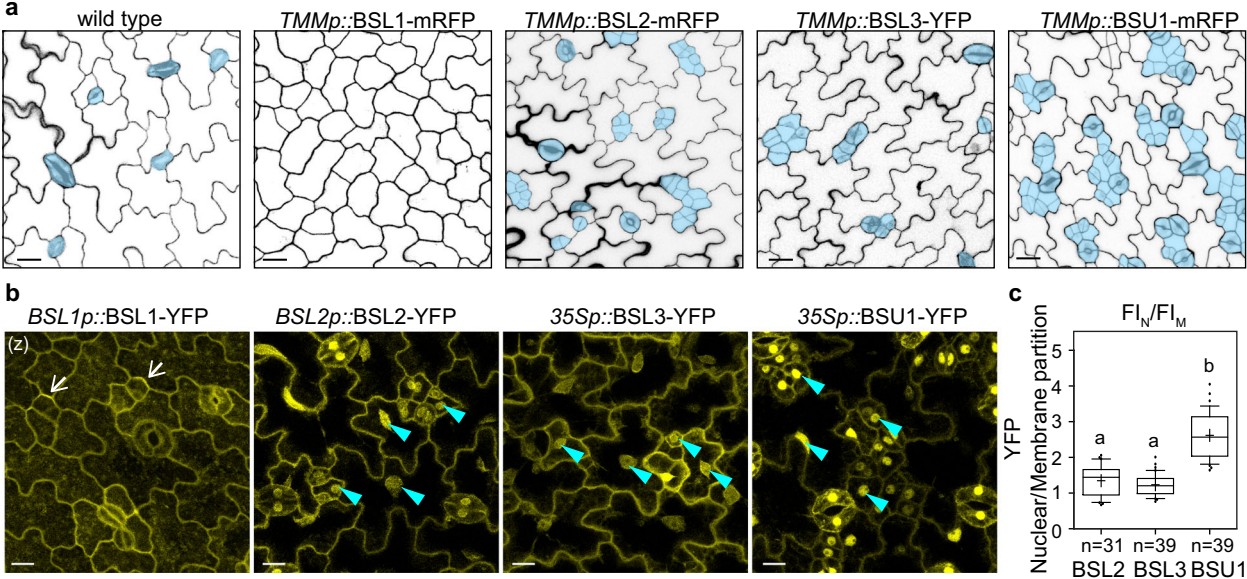

**Fig. 2 Differential overexpression phenotypes and subcellular localization of BSL proteins in stomatal lineage cells. a** Confocal images show stomatal phenotypes induced by overexpression of the *BSL* genes in the stomatal lineage cells (driven by the stomatal lineage *TMM* promoter) in 5-dpg adaxial cotyledon epidermis. Cell outlines were visualized with propidium iodide (PI)-staining. Stomatal lineage and guard cells are manually traced (blue). Data represent results of three independent experiments. Scale bars, 20 μm. **b** Subcellular localization of the BSL proteins (yellow) in stomatal lineage cells in 3-dpg *Arabidopsis* adaxial cotyledon epidermis. Note, BSL1 is mainly cytoplasmic, close to the cell periphery and polarized (arrows); BSL2, BSL3, and BSU1 are cytoplasmic and nuclear (cyan arrowheads). Data represent results of three independent experiments. Scale bars, 10 μm. (z), all images are z-stacked. **c** Quantification of nuclear/membrane (N/M) partition of BSL–YFP by measuring fluorescence intensity (FI) of z-stacked images. Box plots show the first and third quartiles, split by the median (line) and mean (cross). *n*, number of stomatal lineage cells used for quantification. Letters were assigned by ANOVA and Tukey's multiple comparison test ($P \leq 0.01$). Exact *P* values are 0.2208 for BSL3 *vs.* BSL2 and $1.69335^{e-12}$ for BSU1 *vs.* BSL2, respectively, by unpaired *t*-test.

adding an N-terminal myristoylation site to tether the target protein to the PM[17], or by adding a nuclear localization sequence (nls) to target protein to the nucleus (Fig. 3b). Because BSL1 was not present in the nucleus, the other three family members, BSL2, BSL3, and BSU1, were included to be overexpressed in the stomatal lineage cells (driven by the *TMM* promoter, transcript levels examined in Supplementary Fig. 5a). Examination of protein subcellular localization in transgenic plants showed that most of the engineered myr- or nls-BSL2/BSL3/BSU1 proteins exhibited the expected patterns, except that myr-BSU1 showed distribution both at the PM and in the nucleus (Fig. 3a, c, Supplementary Fig. 5b). Examination of stomatal phenotypes revealed an interesting trend that myr-tagged BSL proteins strongly suppressed stomatal production, whereas nls-tagged BSL proteins strongly promoted stomatal production (Fig. 3a, c). This rule is evidently observed for plants expressing myr-BSL2/BSL3 and nls-tagged BSL2/BSL3/BSU1 but not quite applicable to the PM/cytoplasmic/nuclear myr-BSU1-RFP. BSU1 was previously reported to mainly enriched in the nucleus[16]. Overexpression of myr-BSU1 (both PM and nuclear), when compared with the PM-only myr-BSL2/BSL3, was less capable of suppressing stomatal overproduction. On the other hand, when compared with the native, nuclear BSU1, myr-BSU1 was less capable of promoting stomatal production (Fig. 3a, Supplementary Fig. 3a). The mild effect of myr-BSU1 was likely due to its original nuclear retention greatly alleviated by the myr-mediated membrane association. Taken together, the results indicated that the BSL2, BSL3, and BSU1 proteins likely suppress stomatal formation at the PM/cytoplasm but promote stomatal formation in the nucleus.

**Genetic positioning of BSL function at the plasma membrane (PM).** Our previous work had demonstrated that at the PM, the founding member BSL1 relied on its phosphatase activity to directly activate the kinase activity of YDA and jointly release the BIN2-mediated inhibition on YDA, thereby MAPK signaling is elevated in the stomatal lineage cells (Fig. 4a)[15]. Here, in our assays, overexpression of myr-BSL2/BSL3 in the stomatal lineage cells recapitulated the stomata-less phenotype of BSL1 overexpression plants[15] (Figs. 3a, 4b), presumably through the same regulations on YDA and BIN2. Indeed, in the absence of *YDA* or *MPK3/6*, myr-BSL2 failed to suppress stomatal formation, and the *yda* or *mpk3;mpk6* mutant phenotypes were epistatic (Fig. 4c, d, Supplementary Fig. 6a), indicating that the PM function of BSL2 just as BSL1 requires the YDA MAPK signaling pathway. Also, in the absence of three GSK3-like BIN2 and BIN2-like kinases (BIL1/BIL2), the effect of myr-BSL2-mRFP was greatly alleviated in *bin2-3;bil1;bil2* mutants (Fig. 4e, quantification in Supplementary Fig. 6a), suggesting that the BIN2 kinases are also downstream of myr-BSL2. Thus, by using myr-BSL2 as a tool, our results suggested that the PM function of the other BSL members (BSL2/BSL3/BSU1), just as BSL1, requires the YDA MAPK and BIN2 pathways to suppress stomatal differentiation.

**Genetic positioning of BSL function in the nucleus.** We were intrigued by the phenotype of BSL2/BSL3/BSU1-nls and their positive regulation in stomatal development, which strikingly contrasts to their negative regulation at the PM. Because BSL2 showed strong functions at both locations, we used BSL2 as a representative member to genetically characterize the nuclear function of BSL2/BSL3/BSU1. To determine the molecular basis for these BSL proteins to promote stomatal production in the nucleus, we considered the key stomatal regulators, including the MAPK components (YDA, MKK4/5 and MPK3/6), the GSK3-like BIN2 kinases, and the transcription factors SPCH and ICE1/

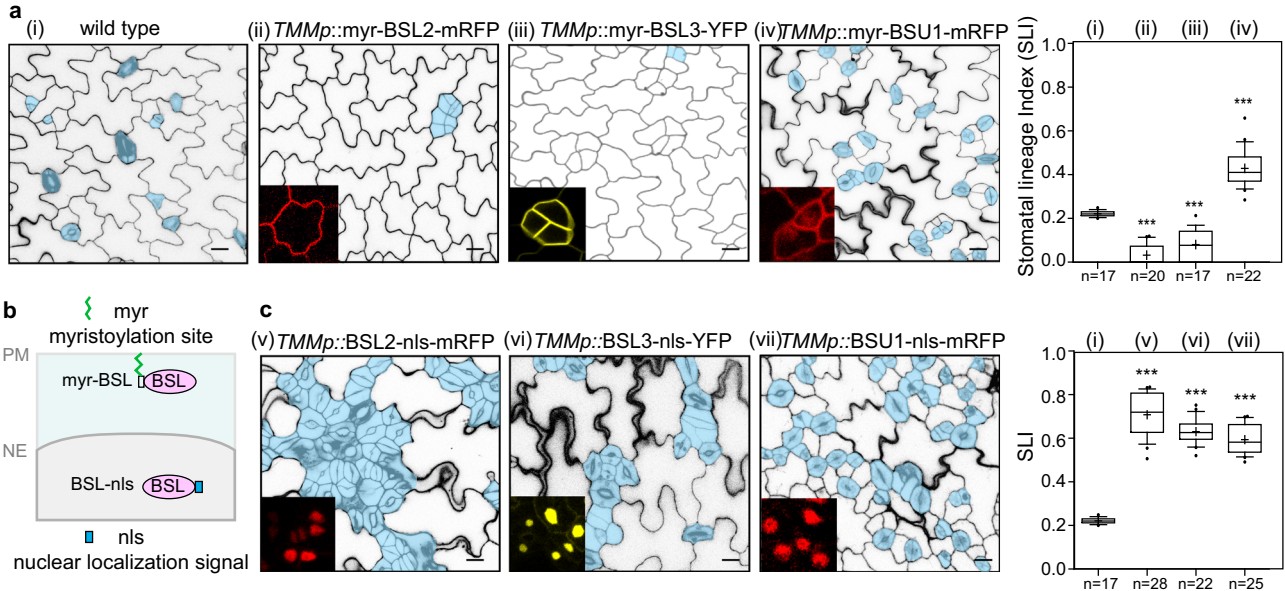

**Fig. 3 Opposite regulation of BSL proteins at the PM vs. in the nucleus. a** Stomatal phenotypes in 5-dpg adaxial cotyledon epidermis overexpressing myristoylated (myr) BSL proteins (driven by the *TMM* promoter). Stomatal lineage cells are manually traced (blue). Insets (red and yellow) show protein localization in stomatal lineage cells. Note, myr-BSL2/BSL3 were solely found at the plasma membrane and myr-BSU1 was localized to the PM, cytoplasm, and nucleus. Each representative confocal images were selected from at least 10 individual cotyledons. Scale bar, 20 μm. **b** Schematic diagram depicts engineered BSL proteins for functional testing at the PM (myr-BSL) or in the nucleus (BSL-nls). PM, plasma membrane; NE, nuclear envelope. **c** Stomatal phenotypes in 5-dpg adaxial cotyledon epidermis overexpressing nuclear BSL-nls proteins (driven by the *TMM* promoter). Insets (red and yellow) show protein localization in the nucleus. Box plots in **a** and **c** show the first and third quartiles, split by the median (line) and mean (cross). *n*, number of biologically independent cotyledons from two homozygous transgenic lines (T3). Statistical analysis was performed to compare with the wild-type values. Unpaired *t*-test, ***$P < 0.0001$. Exact *P* values are $6.65262^{e-15}$ for i *vs.* ii, $1.19704^{e-7}$ for i *vs.* iii, $9.91101^{e-11}$ for i *vs.* iv, $6.4315^{e-21}$ for i *vs.* v, $1.70435^{e-21}$ for i *vs.* vi, $5.90558^{e-21}$ for i *vs.* vii.

SCRMs[5,18–20]. Particularly in the nucleus, both MAPKs and BIN2 kinases may phosphorylate SPCH for degradation[9,20], and MAPKs also regulate the degradation of ICE1/SCRMs[10,21]. To test the genetic interaction with the MAPK pathway, first, we introduced stomata-inducing BSL2-nls into transgenic plants overexpressing one of the constitutively active (CA) MAPK components[4,5,22], such as *CAyda* (driven by the *YDA* promoter), *CAmkk5*, or *CAmpk6* (both driven by the *TMM* promoter), all of which generated a stomata-less phenotype (Fig. 4f–h, top). We found that, strikingly, the overexpression of BSL2-nls was able to recover stomatal production in *CAyda* and *CAmkk5*, but not at all in *CAmpk6* plants (Fig. 4f–h, bottom, quantification in Fig. 4k, transcript levels or genotype examined in Supplementary Fig. 6b, c). The results suggested that BSL2-nls acts downstream or in parallel with MKK4/5 but likely upstream of MPK6 and that the positive regulation of BSL in the nucleus requires MAPK signaling in stomatal development.

Next, we analyzed the relationship between nuclear BSLs and BIN2/BILs. The loss-of-function *bin2-3;bil1;bil2* mutant, due to the defective regulation on YDA and SPCH[12,20], produces fewer stomatal lineage cells in the cotyledons. Interestingly, over-expression of BSL2-nls induced stomatal overproduction in *bin2-3;bil1;bil2* but to a much lesser extent compared to in the wild-type background (Fig. 4i and quantification in Fig. 4k). The results suggested that the effect of BSL-nls at least partially requires the BIN2 kinases or that BLS-nls and BIN2 may act in parallel. Similar results were obtained that the BIN2 kinase inhibitor, bikini, greatly alleviated the effect of BSL2-nls over-expression in plants (Supplementary Fig. 6d). Thus, these results consistently suggested that the GSK3-like BIN2 kinases are partially required for the BSL2-nls. Finally, as anticipated, BSL2-nls overexpression failed to induce stomatal formation in the

stomata-less mutants *spch* and *ice1-2;ice2-1*[18,19] (Fig. 4j, Supplementary Fig. 6e), suggesting these transcription factors are absolutely required for the stomata-promoting function of BSL2-nls. Taken together, our genetic analyses suggested the nuclear function of BSL2 can be connected to BIN2, MPK3/6, SPCH, and/or ICE1/SCRM.

**BSL2 directly interacts with MPK6.** Next, we asked whether the nuclear function of BSL2/BSL3/BSU1 can be achieved by direct physical interaction with the candidates we identified above, i.e., BIN2, MPK3/6, or SPCH/SCRMs. In *Arabidopsis*, it was pre-viously reported that BSU1 directly dephosphorylates BIN2 to suppress its kinase activity in BR signaling[14], so that BIN2 was not included in our interaction assays. However, whether the BSL proteins can interact with MPK3/6, SPCH, and ICE1/SCRMs have not yet been investigated. The yeast two-hybrid results showed that none of the four BSL proteins interacted with MKK4, MKK5, SPCH, or ICE1/SCRM (Fig. 5a). However, BSL2/BSL3/BSU1, but not BSL1, interacted with MPK3/6 (Fig. 5a). Further analysis on the subdomains showed that the interaction of BSL2-MPK6 appeared to require the N-terminal half of BSL2 that contains the Kelch domain (Supplementary Fig. 7a, b).

Because the plasma membrane BSL1 activates YDA[15], the potential interaction of BSL2/BSL3/BSU1 with downstream MPK3/6 is intriguing. The physical interactions between BSL2/BSU1 (BSL3 88% identical BSL2) with MPK6 were further verified by in vitro pull-down assays with proteins produced by *Nicotiana benthamiana* leaf cells. Results showed that BSL2-Flag or BSU1-Flag was co-immunoprecipitated with MPK6-YFP, but not with the YFP-only negative controls (Fig. 5b). Thus, these in vitro assays demonstrated that BSL2/BSL3/BSU1 can directly interact with MPK3/MPK6.

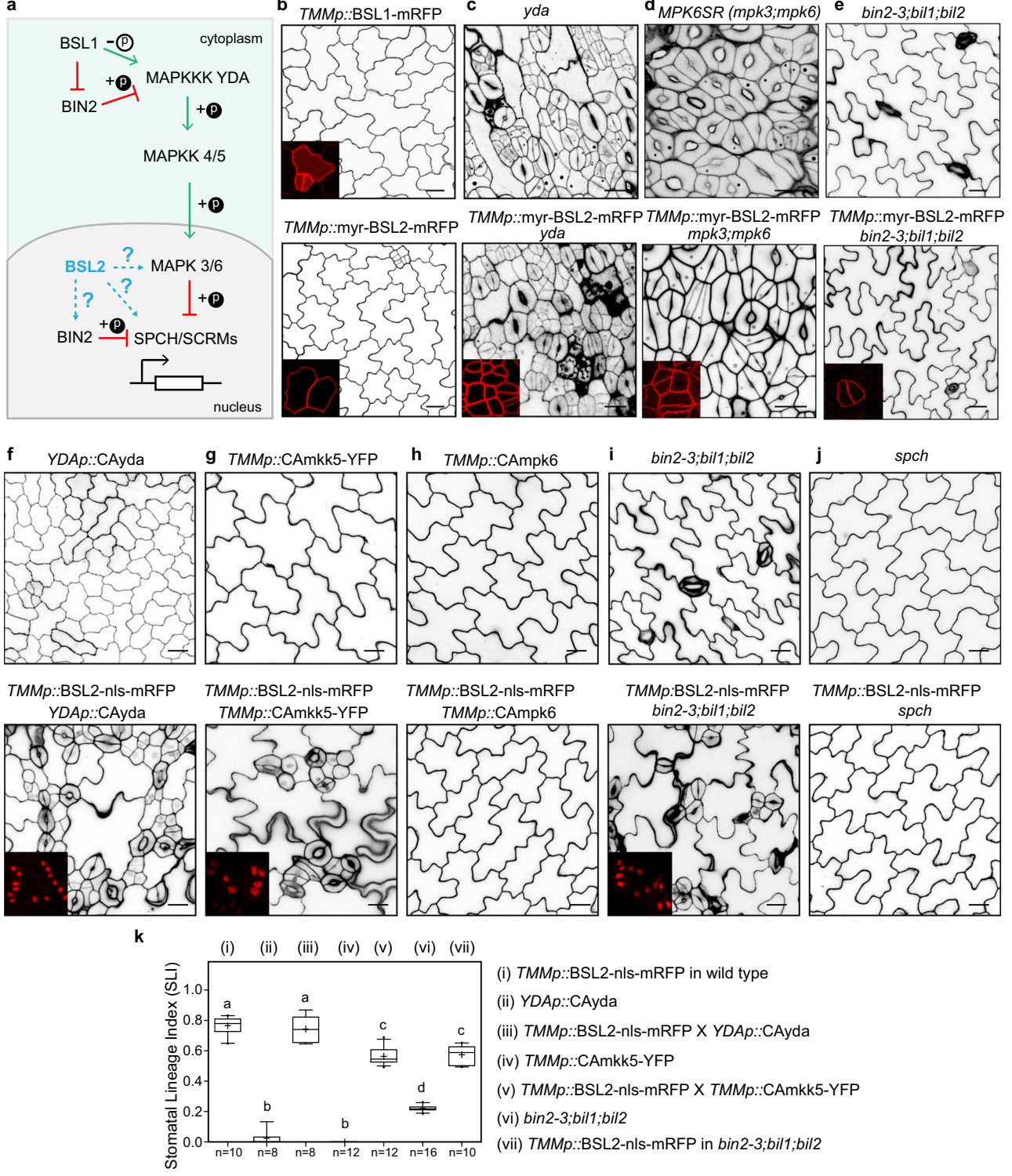

**k**

(i) *TMMp::*BSL2-nls-mRFP in wild type

(ii) *YDAp::*CAyda

(iii) *TMMp::*BSL2-nls-mRFP X *YDAp::*CAyda

(iv) *TMMp::*CAmkk5-YFP

(v) *TMMp::*BSL2-nls-mRFP X *TMMp::*CAmkk5-YFP

(vi) *bin2-3;bil1;bil2*

(vii) *TMMp::*BSL2-nls-mRFP in *bin2-3;bil1;bil2*

Next, we tested protein-protein interaction via the bimolecular fluorescence complementation (BiFC) assays in leaf epidermal cells of *N. benthamiana*. While the negative controls did not produce detectable signals, complemented YFP signals were detected between nYFP-tagged BSL2/BSU1 and cYFP-tagged MPK6, but not with SPCH-cYFP or SCRM-cYFP (Fig. 5c). In addition, the interactions were found to occur in the cytoplasm and nucleus, like those of the BSL2/BSU1 single-protein expressions (Supplementary Fig. 7c). Consistent with the yeast two-hybrid results, BSL2 appeared to interact with MPK6 with its

N-terminal Kelch-containing domain (Supplementary Fig. 7d). Taken together, our results demonstrated physical interactions between BSL2/BSL3/BSU1 and MPK6 both in vitro and in a heterologous plant cell system.

**BSL2 deactivates MPK6.** Given the established phosphatase activity of BSU1 on BIN2 in BR signaling and BSL1 on YDA in stomatal development[14,15], we tested whether BSL2, as a representative nuclear member, may regulate the activity of MPK6. In in vitro kinase assays, when BSL2 was increasingly added into the system, the

**Fig. 4 Nuclear BSL2 functions upstream of MPK6 in stomatal development. a** Graphic diagram describes known (solid lines) and unknown (dashed lines) functional regulations surrounding the linear YDA MAPK signaling module in *Arabidopsis* stomatal development. Arrows and T-bars indicate positive and negative effects, respectively. Phosphorylation modifications are indicated by +p or −p. **b–j** Stomatal phenotypes in 5-dpg adaxial cotyledon epidermis. **b–e** Overexpression of membrane-localized BSL2 (myr-BSL2) was introduced in the wild type (**b**), *yda* (**c**), *mpk3;mpk6* (**d**), or loss-of-function *bin2-3;bil1;bil2* (**e**) mutants (bottom panels) to compare with their respective genetic background (top panels). **f–j** Overexpression of nuclear BSL2 (BSL2-nls) was introduced in the constitutively active variant of YDA (*YDAp*::CAyda, **f**), constitutively active variant of MKK5 (*TMMp*::CAmkk5, **g**), constitutively active variant of MPK6 (*TMMp*::CAmpk6, **h**), *bin2-3;bil1;bil2* triple mutant (**i**), or *spch* loss-of-function mutant (**j**) (bottom panels), to compare with their respective genetic backgrounds (top panels). Cell outlines were stained with PI, images were captured by the confocal microscope and converted to black/white. Insets show protein localization (red) in stomatal lineage cells. Data represent results of three independent experiments. Scale bar, 20 μm. **k** Quantification of stomatal lineage index (SLI) for the designated genotypes. Box plot shows first and third quartiles, median (line) and mean (cross). *n*, number of individual cotyledons. Letters were assigned by ANOVA and Tukey's multiple comparison test ($P \leq 0.01$). Exact $P$ values are $4.7192^{e-15}$ for i *vs.* ii, 0.557 for i *vs.* iii, $3.201^{e-11}$ for i *vs.* iv, $3.90248^{e-7}$ for i *vs.* v, $1.39073^{e-10}$ for i *vs.* vi, $2.32396^{e-6}$ for i *vs.* vii, $6.31636^{e-10}$ for ii *vs.* iii, $2.12926^{e-12}$ for iv *vs.* v, $5.41428e-9$ for vi *vs.* vii, by unpaired *t*-test.

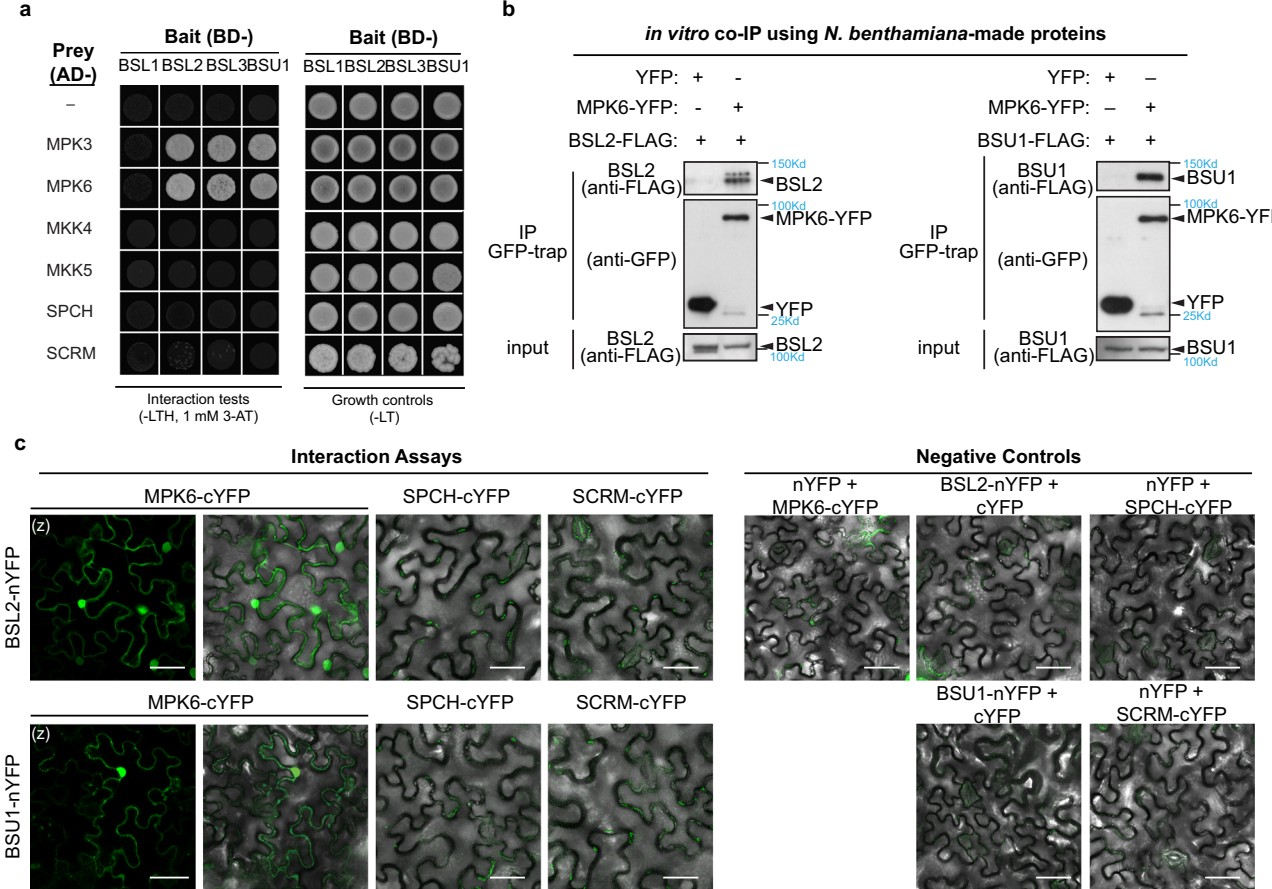

**Fig. 5 BSL2/BSL3/BSU1 interact with MPK6. a** Results of yeast two-hybrid assays show BSL2/BSL3/BSU1 proteins directly interact with MPK3/6. Bait, "BD-" indicates Gal4 DNA-binding domain; Prey, "AD-" indicates Gal4 activation domain. "Growth controls", assays performed using rich media (-Leu-Trp); "Interaction tests", assays performed using synthetic dropout medium (-Leu-Trp-His; 1 mM 3-AT added to suppress bait auto-activation). **b** Co-IP assays using purified fusion proteins produced by *N. benthamiana* leaves show physical association of BSL2-FLAG (left) or BSU1-FLAG (right) with MPK6-YFP. YFP (left lane, - control) or MPK6-YFP (right lane) was used as bait to bind to the GFP-Trap agarose. Immunoprecipitated proteins were detected by anti-FLAG. Data represent results of three biological repeats. **c** Results of bimolecular fluorescence complementation (BiFC) assays in *N. benthamiana* leaf epidermal cells show BSL2 and BSU1 interact with MPK6 in plant cells. YFP signals (green) indicate protein-protein interactions. No interactions were detected between BSL2/BSU1 and the transcription factors SPCH/SCRM. nYFP, N-terminal YFP; cYFP, C-terminal YFP. Data represent results of three independent experiments. Scale, 50 μm. (z), z-staked confocal images.

MPK6 phosphorylation levels were gradually reduced (Fig. 6a). Importantly, the addition of a blend of phosphatase inhibitors (PhosSTOP) abolished the effect of BSL2 (Fig. 6a). Consistent results were obtained when we performed the in vitro kinase assays using myelin basic protein (MBP) as a substrate of MPK6 (Supplementary Fig. 8a). Thus, the results suggested that the kinase activity of MPK6 is inhibited by BSL2 in a dosage-dependent manner.

We further tested whether and how in vivo MPK3/6 activity levels might be eventually altered by the spatially directed overexpression of BSL2 in *Arabidopsis* plants. By using the p42/p44 antibodies to evaluate phosphorylated levels of MPK3/6 in vivo, the immunoblotting results showed that indeed, overexpression of the nuclear enriched BSL2-nls lowered the levels of activated MPK3/6, whereas the PM-enriched myr-BSL2

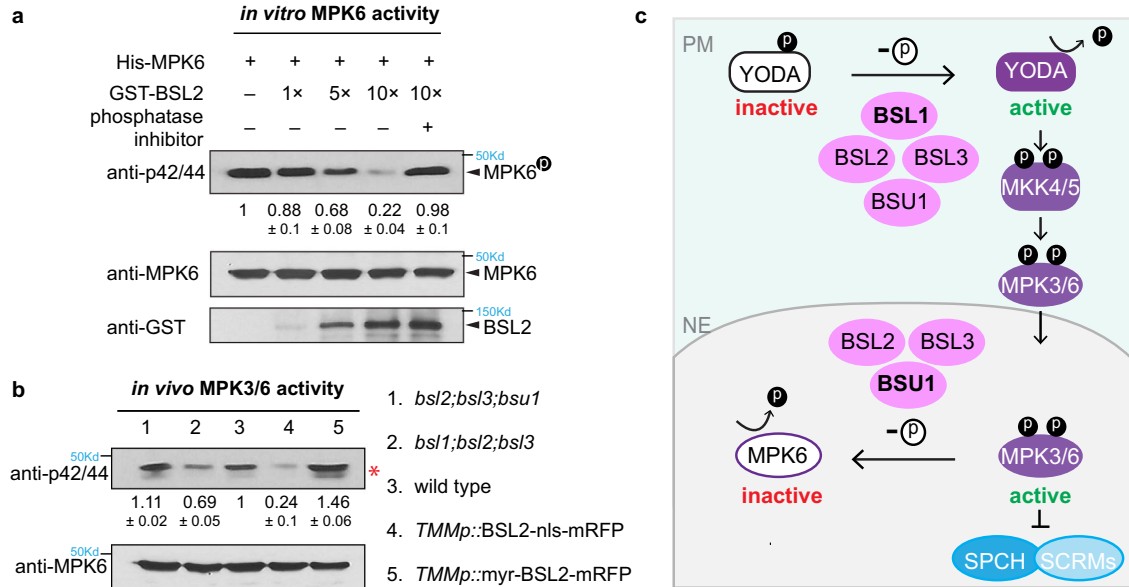

**Fig. 6 Negative regulation of BSL phosphatases on MAPKs in vitro and in vivo. a** Kinase activity of the recombinant MPK6 protein in the presence or absence of BSL2 was examined in vitro. MPK6 and BSL2 were fused histidine-tag (His) and glutathione S-transferase (GST), respectively. phosphorylated MPK6 was detected by immunoblot using anti-phospho-p42/p44 antibody. Protein levels of His-MPK6 or GST-BSL2 were examined using anti-MPK6 or anti-GST antibody, respectively. Numbers indicate relative amount of phosphorylated MPK6 protein of three biological replicates. Data are presented as mean ± SD. **b** Phosphorylated MPK3/6 (activity) levels in 5-dpg *Arabidopsis* seedlings were detected by anti-phospho-p42/p44 (top). Total proteins were extracted, and immunoblotting assays were performed using anti-phospho-p42/p44 (top) and anti-MPK6 (bottom). Numbers are mean ± SD, intensity quantification of phosphorylated MPK6 bands (red asterisk). *n* = three biological replicates. **a**, **b** Representative results of assays performed three times. **c** A working model: A BSL phosphatases-based signaling dichotomy, through spatial compartmentalization of the regulation on distinct components of the linear YODA MAPK pathway, controls stomatal development in *Arabidopsis*. At the cell cortex close to the PM, BSL1 is a predominant regulator, together with the other three BSL phosphatases, activating the MAPKKK YODA to promote MAPK signaling. Activated MPK3/6 molecules phosphorylate the key stomatal fate transcription factors, SPCH and ICE1/SCRMs, for degradation, thereby suppressing stomatal production. In the nucleus, BSU1 plays a primary role, together with BSL2 and BSL3, deactivating MPK3/6, resulting in stabilized SPCH and ICE1/SCRMs, thereby promoting stomatal production.

induced elevated levels of activated MPK3/6 in plants (Fig. 6b). In addition, we found that mutating the three major nuclear *BSL* members, i.e., *BSL2*, *BSL3*, and *BSU1*, resulted in elevated levels of activated MPK3/6 in *bsl2;bsl3;bsu1* plants, whereas mutating *BSL1*, the predominant regulator at the PM, in combination with *bsl2;bsl3* mutations converted the levels of activated MPK3/6 to the lower side in *bsl1;bsl2;bsl3* plants (Fig. 6b). The negative regulation of nuclear-targeted BSL2/BSL3/BSU1 on MPK3/6 was further supported by examining the protein levels of SPCH (*SPCHp*::SPCH-CFP) relative to its transcription levels (*SPCHp*::nucYFP) in vivo. As a MPK3/6 substrate, the protein abundance of SPCH was downregulated by MPK3/6 in wild-type plants (fewer CFP-positive cells than YFP-positive cells at the same developmental stage). But this downregulation was greatly alleviated by BSL2-nls or BSU1 overexpression (Supplementary Fig. 8b–e). Taken together, our results suggested that the nuclear function of BSL proteins may directly deactivate MPK3/6 to promote SPCH protein abundance, thereby stomatal production in *Arabidopsis* (Fig. 6c).

## Discussion
In this study, our genetic and cell biological analyses reveal a functional dichotomy of the four Kelch-domain containing BSL phosphatases in the regulation of *Arabidopsis* stomatal development (Fig. 6c and Supplementary Fig. 8f). Through phenotypic analyses of the loss-of-function mutants, we identified an opposing contribution from differentially combined family members, i.e., *BSL2* and *BSL3* can be either negative or positive regulators of stomatal production depending on whether they are combined with *BSL1* or *BSU1*, respectively. When combined with *BSL1*, *BSL2/BSL3* confer a negative regulation, whereas when

combined with *BSU1*, *BSL2/BSL3* confer a positive regulation (Fig. 1, Supplementary Fig. 2). This functional paradox can be explained by the spatial compartmentation of the BSL-mediated signaling events at the subcellular level (Fig. 6c). At the cell cortex, predominantly driven by BSL1, all four BSL proteins contribute to the activation of the YDA MAPK signaling pathway, which triggers the phosphorylation and degradation of SPCH and ICE1/SCRMs, resulting in the suppression of stomatal production. Therefore, in *bsl-q* mutants, the loss of activation on YDA kinase results in the loss of MAPK signaling, leading to elevated SPCH abundance and stomatal overproduction. In the nucleus, the predominant contributor BSU1, together with BSL2/BSL3, deactivates MPK3/6, thereby stabilizing SPCH and ICE1/SCRMs to promote stomatal production. Therefore, in *bsl2;bsl3;bsu1* triple mutants, the presence of BSL1 is sufficient to activate YDA kinase but the absence of the nuclear inhibitors (BSL2/BSL3/BSU1) results in elevated MAPK signaling, leading to reduced SPCH abundance and stomatal inhibition. Thus, our study identified a signaling dichotomy of the BSL phosphatases, which can positively or negatively regulate a specific tier of the YDA MAPK cascade to control stomatal development (Fig. 6c). Hence, by shifting the spatial distribution of BSL2 and BSL3 that are dually localized to the nucleus and the PM/cytoplasm, the meristemoid and SLGC can decide its stomatal or non-stomatal cell fate, thereby regulating the density of guard cells in the leaf epidermis.

The BSL proteins belong to the family of Protein Phosphatase with Kelch-like domains (PPKL). While BSL1, BSL2, and BSL3 are highly conserved and universally present in all green plants, BSU1-type genes are exclusively found in Brassicaceae[23]. The four BSL members were initially thought to function redundantly in

BR signaling[14,24], but more recent genetic data suggest that BSL1/BSL2/BSL3 and BSU1 have contrasting effects on BR signaling[23], The mechanism for this functional divergence remains unknown. In our study, phenotypic comparison of the various *bsl* mutants also supported contrasting roles of the BSL proteins in stomatal production, particularly for *BSL1 vs. BSU1* (Supplementary Fig. 2b). Apart from many other reasons, such as cell-type-specific expression, conferring the differential functions of individual *BSL* members, our studies here showed that spatial compartmentation of signaling events, centering on the components of a MAPK cascade, provides a powerful mechanism for the BSL proteins to both positively and negatively modulate stomatal formation in *Arabidopsis*.

In our working model, all four BSL members at the PM contribute to the negative regulation of stomatal differentiation (Fig. 6c). This was based on the facts that (1) gradually more severe stomatal overproduction phenotypes were observed in the double, triple, and quadruple mutants, and (2) overexpression of membrane-associated/enriched BSL1 and myr-BSL2/BSL3/BSU1 were all capable of suppressing stomatal production (Fig. 3a). The previous studies by Kim et al.[16] and Guo et al.[15] demonstrated that BSL2, BSL3, and BSU1 are cytoplasmic and nuclear, whereas BSL1 is exclusively cytoplasmic and close to the PM. A specific amino acid sequence (GTLDE) was later identified to direct the BSL1 PM/cytoplasmic retention by an unknown mechanism[16]. In the stomatal lineage cells, direct physical interactions between the polarity protein BASL and BSL1/BSL2/BSL3 may recruit these BSLs to the PM polarity site. BSL1 was demonstrated to directly interact with and dephosphorylate YDA, leading to YDA activation[15]. Because myr-BSL2 promoted the YDA pathway, and because *yda* and *mpk3;mpk6* were epistatic to myr-BSL2 (Fig. 4c, d), we suggest that, like BSL1, the other BSL phosphatases also participate in the direct dephosphorylation of YDA, resulting in elevated YDA activity and MAPK signaling in plants.

Then, what activates the BSL phosphatases before they can activate YDA? In embryo development, the membrane-associated pseudokinase SHORT SUSPENSOR (SSP) was found to constitutively activate YDA by physical binding[25,26], a process likely facilitated by the Gβ protein AGB1[27]. *BRASSINOSTEROID SIGNALING KINASE 1* (*BSK1*) and *BSK2*, two close homologs of *SSP*, were recently identified to participate in YDA-dependent signaling in stomatal development[26]. In addition, in BR signaling, the activation of BSU1 is triggered by BSK1 that is phosphorylated by the cell-surface brassinosteroid receptor BRASSINOSTEROID INSENSITIVE 1 (BRI1)[12,14]. Thus, the outstanding question is whether the BSL phosphatases are activated by BSKs prior to they can activate the YDA MAPK signaling. it is possible that the BSL family members function between BSKs and YDA to relay the Brassinosteriods or ligand-receptor triggered signaling to the MAPK cascade.

While YDA is predominantly localized in the cytoplasm and tightly associated with the PM[28], both MKK4/5 and MPK3/6 are broadly distributed in the cytoplasm and the nucleus[29]. It is not uncommon that more than one tier of the MAPK cascade is targeted by one regulator. For example, the conserved GSK3-like BIN2 kinase was first determined to phosphorylate and inhibit YDA in BR-regulated stomatal development[12], and later found to catalyze the phosphorylation of the MAPK Kinase 4, reducing MKK4 activity against its substrate MPK6[13]. However, our demonstration of opposing functions for the BSL proteins to regulate multiple tiers of a MAPK cascade is striking. In this study, we demonstrated that, besides the previously reported BSL1 interaction and inhibition of the upstream MAPKKK YDA[15], BSL2/BSL3/BSU1 interact and deactivate downstream MPK6 signaling in vitro and in vivo (Figs. 5, 6). No direct physical interactions between the BSL proteins with MAPKKs were identified yet.

Conventional MAPKs are known to be activated by MAPKKs on the conserved catalytic loop Thr-X-Tyr (TxY)[30] and deactivated by protein phosphatase-mediated dephosphorylation[31]. Here, we show that BSL2/BSL3/BSU1 bind to MPK3/6 in vitro, deactivate MPK3/6 signaling, and promote the protein stability of key transcription factor SPCH in vivo (Figs. 5, 6, Supplementary Fig. 8). The BSL proteins are members of Ser/Thr phosphatases characterized by a C-terminal catalytic domain. The phosphatase activity is not only crucial for BSL1 to activate YDA and regulate BIN2[14,15], we found it is also essential for BSL2 to deactivate MPK6 (Fig. 6). It is likely that BSL2/BSL3/BSU1 dephosphorylate the Thr site of the TxY motif to counteract the MAPKK-mediated activation of MPK3/6. Indeed, we observed moderately reduced phenotype of BSL2-nls when co-expressed with the constitutively active MKK5 in *Arabidopsis* (Fig. 4g), suggesting an antagonistic regulation of BSL phosphatases and MAPKKs against MPK3/6 in stomatal development. Furthermore, the BSL phosphatases contain an N-terminal Kelch repeats that form β-propeller structures, capable of mediating protein-protein interactions[32]. Our interaction assays did detect direct physical interaction between the N-terminal half of BSL2 and MPK6 (Supplementary Fig. 7). What other regulators bind to the Kelch domain and how they regulate the BSL activities are important questions to pursue in the future.

The striking positive regulation of BSL2/BSL3/BSU1 in stomatal development appeared to occur predominantly in the nucleus (Fig. 3), though BSU1 was previously shown to control BR signaling mainly in the cytoplasm[14,33]. Clarification of the mechanisms controlling the nuclear/cytoplasmic shutting of BSU1 and BSL2/BSL3 remains challenging because the BSL proteins do not contain a nuclear localization signal, thus their nuclear retention is probably regulated by interacting with substrates or scaffold proteins. Interestingly, BIN2 as one of the substrates of BSU1 was shown to reside both in the nucleus and the cytosol[15]. In stomatal development, BIN2 is temporally recruited to the cortical polarity complex, while the nuclear retention of BIN2 is promoted by the PM-enriched BSL1 phosphatase[15], as well as a negative regulatory loop with the scaffold protein POLAR at the cell cortex[34]. In the nucleus, BIN2 and MPK3/6 promote SPCH degradation via direct phosphorylation[9,20]. Therefore, the nuclear enrichment of substrates, such as BIN2 and MPK3/6, can be one of the reasons that promotes the nuclear function of BSL2/BSL3/BSU1 (Supplementary Fig. 8f).

One key question related to the signaling dichotomy of the BSL phosphatase is how opposite signaling events in the regulation of the linear YDA MPK3/6 pathway are differentially triggered and modulated in plant development. The manifested stomatal defects in the loss-of-function *bsl*-quad mutants (severely over proliferative guard cells) suggest the negative regulation is default for the four *BSL* genes in this developmental context. Consistently, the primary drivers of the negative regulation, *BSL1/BSL2/BSL3*, are deeply conserved and essential for many developmental programs, including early endosperm development[23]. As discussed above, the activation of BSL phosphatases at the cell periphery might rely on the partnership with BSKs and/or other regulators. The predominant nuclear function of BSU1 was suggested to evolutionarily split from the other family members and might have undergone a process of neofunctionalization[35]. What signals trigger the nuclear function of BSL2/BSL3/BSU1 are unknown but BSU1 in BR signaling was found to be activated in the cytoplasm prior to its activity of deactivating BIN2 in the nucleus[36]. As stomatal development can be heavily influenced by environmental changes[2,3], it is reasonable to suspect that hormonal, such as BR, and/or other environmental cues may trigger the shift of the BSL signaling function between the PM/cytoplasm and the nucleus to impact on stomatal production. The next step

towards understanding the opposing functions of BSL members in the regulation of stomatal development, probably applicable to other developmental processes as well, is to dissect out the spatially segregated protein networks that associate with BSL at the cell periphery and in the nucleus, respectively. Regulators responding to environmental changes would be great candidates for their possible participation in the regulation of BSL nuclear/cytoplasmic shuttling.

## Methods

**Plant materials**. The *Arabidopsis thaliana* ecotype Columbia (Col-0) was used as the wild type. All mutants are in the wild type Col-0 background except for *bin2-3;bil1;bil2* in the Wassilewskija (Ws-0) background. The *bsl* T-DNA insertional mutants were obtained from the Arabidopsis Biological Resource Center (ABRC), including *bsl1* (SALK_051383), *bsl2* (SALK_055335), *bsl3* (SALK_072437), and *bsu1* (SALK_030721). These null alleles were used for generating higher-order mutants and phenotypic analysis. Mutants and transgenic *Arabidopsis* lines reported previously were: *spch-3*, SPCHp::SPCH-CFP;*spch-3*[37], SPCHp::nucYFP[38], *ice1-2;ice2-1*[39], *yda-3* (Salk_105078)[28], YDAp::CAyda[40], *bsl-quad* (null alleles of *bsu1*, *bsl1* combined with RNAi-mediated silencing of *BSL2* and *BSL3*), *bsl1a-miRNA-BSL2;3*, *amiRNA-BSL2;3*, and *bin2-3;bil1;bil2*[12], *mpk3;mpk6-6SR*[41], BSL1p::BSL1-YFP, BSL2p::BSL2-YFP[15], 35Sp::BSL3-YFP, 35Sp::BSU1-YFP[12]. Primers for genotyping and quantitative real-time PCR are listed in Supplementary Table 1.

**Plant growth conditions**. In general, *Arabidopsis* seeds were surface sterilized with 10% bleach and grown on half-strength Murashige and Skoog (MS) basal medium plates or in soil with 16-h light/8-h dark cycles at 22 °C. Wild-type *N. benthamiana* plants were grown under 14-h light and 10-h darkness at 25 °C.

**Plasmid construction**. The Gateway cloning technology (Invitrogen) was used for most DNA manipulations unless otherwise specified. For molecular cloning, the coding DNA sequences (CDS) of *BSL1*, *BSL2*, *BSL3*, *BSU1*, *myr-BSL2/3/U1*, *BSL2/3/U1-nls*, *MPK6*, and *CAmkk5* were cloned into pENTR/D/TOPO vectors (Invitrogen). The *MPK6*[D218GE222A](CAmpk6) were obtained by site-directed mutagenesis of *MPK6* CDS in pENTR/D/TOPO by QuikChange II Site-Directed Mutagenesis Kit (Agilent). The *TMM* promoter sequences can be found in ref. [7] and were subcloned into pDONR-P4-P1R (courtesy of Dr. Diego Wengier). Double LR recombination reactions using LR Clonase II (Invitrogen) were performed to integrate pENTR/D containing CDS of the gene-of-interest and pDONR-promoter into the R4pGWB vectors[42]. For transient protein expression in *N. benthamiana*, the pENTR/D vectors containing the coding sequences of *MPK6, SPCH, SCRM, BSL1, BSL2, BSL3, BSU1*, N-terminal, or C-terminal of *BSL2* were recombined into the pH35GC/Y[43] or pGWB to generate 35Sp::BSLf-CFP, 35Sp::MPK6/SPCH/SCRM-CFP, or 35Sp::BSL2/BSU1-FLAG. The pXNGW and/or pXCGW vectors were used for recombination reactions to generate the BiFC constructs. The resulted binary vectors mentioned above were confirmed by restriction enzyme digestion and DNA sequencing, then transferred into *Agrobacterium tumefaciens* strains GV3101 for *Arabidopsis* transformation and/or *N. benthamiana* leaf infiltration.

**Confocal imaging and image processing**. Confocal images were acquired by a Leica TCS SP5 II microscope. The excitation/emission spectra for various fluorescent proteins are: CFP, 458 nm/480–500 nm; GFP, 488 nm/501–528 nm; YFP, 514 nm/520–540 nm; mRFP, 594 nm/600–620 nm, and propidium iodide (PI), 594 nm/591–636 nm. Images were taken from similar central areas in the adaxial side of developing cotyledons of *Arabidopsis* seedlings or *N. benthamiana* leaves. Cells outlines in *Arabidopsis* were visualized by propidium iodide (PI, Invitrogen) staining. All imaging processing was performed with Fiji (Image J) software (http://fiji.sc/Fiji). Whenever possible, z-stacked images were obtained. Quantifications and statistical analyses were performed using Fiji and GraphPad Prism 5.1, respectively.

**Recombinant protein expression, purification, and in vitro kinase assay**. To express recombinant proteins in *E. coli*, the coding region of *MPK6* was cloned into pET28a vector to generate His-tagged MPK6. The coding region of *BSL2* or *MKK5*[DD] were cloned into pGEX-4T-1 vector to generate GST-tagged BSL2 and GST-tagged MKK5[DD]. The primer sequences are listed in Supplementary Table 1.

All constructs were transformed into *E. coli* (BL21 strain) and recombinant protein expression was induced by IPTG (0.5 mM) at 16 °C for 16 h. Bacterial cells were harvested and lysed by sonication in lysis buffer (50 mM Tris-HCl pH 7.4, 150 mM NaCl, 0.5 mM EDTA, and 1 mM PMSF). GST-tagged proteins were purified with Pierce™ Glutathione Superflow Agarose (Thermo Scientific™), and His-tagged proteins were purified with Ni-NTA Agarose (Qiagen) according to the manufacturer's instructions.

Purified proteins were used for in vitro kinase assay. His-MPK6 proteins were immobilized on Ni-NTA Agarose, which were then incubated with constitutively

active MKK5(MKK5[DD]), in the presence of ATP. The column-bound activated His-MPK6 was washed with buffer to remove any residual MKK5[DD]. Pre-phosphorylated MPK6 was incubated without or with gradually increased amount of GST-BSL2, in the presence or absence of a mixer of phosphatase inhibitors, PhosSTOP (Sigma-Aldrich). The reaction mixture was subjected to standard western blotting analysis and immunodetection with anti-phospho-p42/p44 antibody (1:1000, Cell Signaling Technology).

When MBP were used as a substrate for in vitro kinase assay. Pre-phosphorylated MPK6 was incubated with MBP, without or with gradually increased amount of GST-BSL2, in the presence or absence of a mixer of phosphatase inhibitors, PhosSTOP (Sigma-Aldrich). After adding p-nitrobenzyl mesylate (PNBM) for alkylating the potential thiophosphoryl group on MBP, the reaction mixture was subjected to standard western blotting analysis and immunodetection with anti-Thiophosphate ester antibody (Abcam, #ab92570).

**Protein transient expression in *Nicotiana benthamiana* leaf epidermis and image analysis**. *Agrobacterium* strains GV3101 harboring the constructs-of-interest in 10 ml of LB medium with appropriate antibiotics were cultured overnight. Bacterial cells were collected at $5000 \times g$ for 10 min and resuspended in 10 mM MgCl₂. Cell culture and p19 were mixed to reach an $OD_{600} = 0.5$ for each line and co-infiltrated into abaxial side leaves of 4-week-old *N. benthamiana* plants, as described previously[28]. Co-infiltrated leaves were checked by confocal microscopy 3–4 days post infiltration.

**Co-immunoprecipitation (co-IP) assay using proteins produced by *N. benthamiana***. To test physical association between BSL2 or BSU1 with MPK6 in plant cells, total cell proteins were extracted from *N. benthamiana* leaves that transiently expressed the combinations of 35Sp::YFP or 35Sp::MPK6-YFP with 35Sp::BSL2-FLAG or 35Sp::BSU1-FLAG. *N. benthamiana* leaves were collected and ground in liquid nitrogen with the protein extraction buffer (100 mM Tris-HCl pH 7.5, 5 mM EDTA, 5 mM EGTA, 1 mM Na₃VO₄, 50 mM NaF, 50 mM b-glycerophosphate, 10 mM DTT, 1 mM phenylmethylsulfonyl fluoride, 5% (v/v) glycerol, 0.5% (v/v) Triton X-100, and 1% (v/v) protease inhibitor cocktail (Sigma-Aldrich, P 9599)). Protein extracts were centrifuged at $18,000 \times g$ at 4 °C for 30 min and the supernatants were incubated with GFP-Trap Agarose beads (Chromotek) at 4 °C for 3 h. Then, the beads were washed three times with extraction buffer, followed by mixing with 2 × SDS sample buffer and boiling for 5 min. Samples were separated by 10% SDS–PAGE and analyzed by corresponding primary antibodies (Monoclonal ANTI-FLAG M2 antibody produced in mouse, F3165, Sigma-Aldrich; anti-GFP Antibody, Roche #11814460001).

**In vivo MAPK activity assay**. Total cell proteins were extracted with extraction buffer (50 mM HEPES pH 7.5, 150 mM NaCl, 5 mM EDTA, 5 mM EGTA, 10 mM DTT, 10 mM Na₃VO₄, 20 mM NaF, 50 mM β-glycerophosphate, 10% glycerol, 1 mM PMSF, protease inhibitor cocktail for plant cell extracts (Sigma-Aldrich, P 9599), 1% (v/v) NP-40) from 5-dpg *Arabidopsis* seedlings of Col-0, *bsl2;bsl3;bsu1, bsl1;bsl2;bsl3*, TMMp::myr-BSL2-mRFP, and TMMp::BSL2-nls-mRFP. The extracted total cell proteins were resolved on 10% SDS–PAGE, followed by Immunoblot with the primary antibody against phosphor-p42/44 MAP kinase (1:1000, Cell Signaling Technology). Equal loading was indicated by immunoblot analysis with anti-MPK6 antibody.

**Yeast two-hybrid assay**. Full-length coding sequence of *BSL1, BSL2, BSL3, BSU1*, N-terminal, or C-terminal of *BSL2* was cloned and inserted into the pGBKT7 vector as bait. The full-length coding sequence of *MPK3/6, MKK4/5* or *SPCH/SCRM* was cloned into the pGADT7 vector. Constructs used for testing the interactions were co-transformed into *Saccharomyces cerevisiae* strain AH109 using EZ-YEAST™ transformation kit (MP Bio-medicals) following the manufacturer's instructions. The positive transformants were selected on the SD/-Leu/-Trp medium. The interactions were tested on the SD/-Leu/-Trp/-His medium with appropriate concentration of 3-amino-1,2,4-triazole (3-AT). Interactions were observed after 3 days of yeast growth at 30 °C.

**Quantitative real-time PCR**. Total RNAs were extracted from 50 mg of seedlings at 3 to 5-dpg with RNeasy Plant Mini Kit (Qiagen). cDNAs were generated by the SuperScript™ III First-Strand Synthesis System (Invitrogen). Transcript levels of *BSL1, BSL2, BSL3*, and *BSU1* were amplified with the primers listed in Supplementary Table 1 and the reactions were set up by SYBR™ Select Master Mix (Thermo Fisher Scientific). *ACTIN 2* was used as an internal control to normalize expression levels. Data are presented as mean ± SD. Quantitative real-time PCRs were performed by the Stepone real-time PCR system (Applied Biosystems).

**Stomatal quantification**. In general, to examine stomatal phenotypes in development, 5-dpg (days post germination) cotyledons were stained with PI (Invitrogen) to capture images from similar central regions of adaxial cotyledons. Images were captured by the EC Plan-Neofluar (20×/0.5) lenses on a Carl Zeiss Axio Scope A1 fluorescence microscope equipped with a Progress MF CCD camera (Jenoptik). Typically, 12–20 individual seedlings were picked from each mutant or two

representative T2 transgenic lines out of >12 independent transgenic events. Confocal images shown in the figures were false colored with brightness/contrast adjusted by Fiji.

To quantify stomatal phenotypes, the epidermal cells were categorized by size and shape into three groups: guard cells (pairs of kidney-shaped), stomatal lineage cells (small dividing cells, including MMCs, Ms, and SLGCs), and pavement cells (puzzle-shaped epidermal cell and enlarged SLGCs with at least one obvious lobe). Quantification for stomatal lineage index (SLI: number of guard cell pairs + stomatal lineage cells over total number of epidermal cells) were calculated by counting cells in an area of 0.385 mm$^2$ with the cell-counter plug-in in Fiji.

To quantify SPCH expression patterns, seedlings expressing *SPCHp*::nucYFP or *SPCHp*::SPCH-CFP in the wild-type, *TMMp*::BSL2-nls-mRFP, or *TMMp*::BSU1-mRFP were stained with PI and images of the adaxial side cotyledon epidermis were captured by Confocal. Numbers of Y/CFP-positive cells and total stomatal lineage cells were counted with Fiji to obtain the Y/CFP-positive ratios.

**Statistics and reproducibility**. All statistical analyses were conducted with GraphPad Prism 5.1 Software. To compare two normally distributed groups, unpaired two-tailed *t*-tests were used. For multiple comparisons between normally distributed groups, one-way ANOVA followed by Tukey's post hoc test were used. For all figures, \*$P < 0.05$, \*\*$P < 0.005$, \*\*\*$P < 0.0001$. To quantify immunoblots, the grid and image counter plug-ins in Image J were used. Numbers of repetitions and replicates for each experiment were indicated in the legends.

**Reporting summary**. Further information on research design is available in the Nature Research Reporting Summary linked to this article.

## Data availability

All data generated or analyzed during this study are included (figures, figure legends, and Supplementary Information files). Supplementary data, including figures and tables, are available for this paper. Source data are provided with this paper.

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

## Acknowledgements

We thank Dr. Dongtao Ren (China Agricultural University) and Dr. Zhiyong Wang (Stanford University) for sharing plant reagents and DNA resources. We thank Dr. Tsuyoshi Nakagawa (Shimane University) for sharing the R4 Gateway Binary Vectors (R4pGWB). We thank the ABRC stock center for providing T-DNA insertional mutants. This research was supported by grants from the National Institute of Health GM109080 and GM131827 to J.D. J.D. is also supported by grants from the National Science Foundation 2049642, 1952823, and 1851907.

## Author contributions

X.G. and J.D. designed the research. X.G. and X.D. performed the experiments. X.G. and J.D. wrote the manuscript.

## Competing interests

The authors declare no competing interests.
