## [Peer Review File · Nature Communications]

Dichotomy of the BSL phosphatase signaling spatially regulates MAPK components in stomatal fate determinationREVIEWER COMMENTS

Reviewer #1 (Remarks to the Author):

In this manuscript from the Dong lab, the authors explored the function of the BSL2, BSL3 and BSU1 phosphatases in stomatal development. The authors previously showed that BSL1, a member of the BSL family, is associated with the polarity complex on the cell membrane and activates the MAPKKK YODA in suppressing stomatal development. Here, using a combination of genetic, microscopic and biochemical approaches, they propose that the other three members of the family can function in the nucleus, and, in contrast to the role of BSL1, promote stomatal development by inhibiting nuclear MPK6.

The finding that the BSL phosphatases exhibit opposing functions in a cell compartment-dependent manner is novel and intriguing, and the authors presented a mechanism for the inhibitory effect in the nucleus. The work also helps dissect the role of individual BSLs in stomatal development. The experiments were generally well designed and executed. There are, however, a few points that I'd like the authors to address.

Major points:

1. Although the proposed inhibitory roles of BSL2 and 3 in stomatal development is mostly clear, the same can't be said for BSU1. The conclusion that BSU1 can suppress stomatal development (e.g. Line 78) is mainly based on the *bsl-q* mutant, which, as the authors stated, contains T-DNA insertions at BSU1 and BSL1 and an amiRNA construct(s) against both BSL2 and 3. Compared with T-DNA mutants, the amiRNA approach has a potential drawback in its off-target effect, i.e., in this case, towards unknown BSL-related genes, and causes artifacts.

Indeed, based on the authors' data of the double and triple T-DNA mutants, mutations in BSU1 have never resulted in higher stomatal lineage index, but rather led to significant drops, which contradict with the *bsl-q* phenotype.

Thus, I suggest the authors to re-examine their data, and clarify the native roles of BSU1. Instead of relying solely on *bsl-q*, it would be ideal to assess the phenotype of T-DNA (or CRISPR)-based quadruple *bsl1/2/3 bsu1* mutants. Examining a single *bsl1* mutant with the same amiRNA against BSL2 and 3 (e.g. by crossing out the *bsu1* in *bsl-q*) and comparing it with *bsl-q* may also help address the effect of both the amiRNA and BSU1. Though, I also think the authors have sufficient genetic data that a careful re-assessment of them may suffice.

Further, I found the section describing the double/triple mutant phenotypes somewhat confusing, and improvements on its presentation would be needed.

2. The in vitro assay that assessed the effect of BSL2 on MPK6 activity (Fig. 6a) used the phosphorylation status of MBP as a readout of MPK6 activity. However, the observed effect on MBP could be due to the BSL2 activity acting on the MBP protein itself. Thus, probing the effect on phosphorylated MPK6 (by anti-p42/44 or phos-tag gel) in the presence of BSL2 would be a superior readout and will better support the authors' claim that BSL2 acts on MPK6 directly.

Further, the amount of the 5x BSL2 reaction seems off in the assay as judged by the anti-GST blot.

3. The manuscript would be strengthened by having some initial investigations on whether the membrane/nuclear distribution of BSL2/3/BSU1 is dynamic or not. Given that BSL2/3 can interact with BASL, quantification of their distribution in stomatal lineage cells at different developmental stages, similar to what the authors have showed for BSL1 in their previous work, would be a reasonable start.

Other points:

- What is the transcript level of BSL2 and 3 in the *bsl-q* mutant compared with WT?

- A few genetic materials and data (e.g. TMMp:BSL1-mRFP and localization of BSL2/3 and BSU1) were first appeared in the authors' last publication on BSL1. The authors should refer to the work when first

presenting them to distinguish the novelties of the current study.

- The loading control (rubisco staining) in Fig. 6b is not clear. Since quantitative data were presented, it'd be better to represent loading by probing control proteins by Western blot.

Reviewer #2 (Remarks to the Author):

Stomata formation is regulated by a cascade of MAP kinases. The manuscript of Guo and Dong describes regulation of this cascade by four BSL phosphatases. Previously, this group showed that one of these phosphatases, BSL1, activates MAPKKK YODA at the plasma membrane. Here, they show that at the plasma membrane all four phosphatases dephosphorylate YODA which activates it. But in the nucleus, three out of the four phosphatases dephosphorylate a MAPK MPK6 which inactivates it. The authors clearly demonstrate that when subcellular localization of these phosphatases changes, their substrate changes too. This is an elegant paper that significantly contributes to our understanding of the regulatory mechanism of MAPK cascades, and it demonstrates how a phosphatase can function as both a positive and negative regulator of a signaling pathway. In addition, it provides a great example of how the nature of an enzyme's substrates can depend on the enzyme's subcellular localization. The data are comprehensive, of high quality and support the conclusions. The writing is clear and logical. Minor suggestions:

1. Figure 1B. Images (d) and (e) show different phenotypes. There is much more stomata clustering in (e) compared to (d). However, on the graph, the phenotype of these two mutants has no statistical difference. Are these images representative? Is SLI a good measurement of these phenotypes?
2. The use of asterisks in Fig 1B is confusing. A compact letter display would be better for visualization in this figure.

REVIEWER COMMENTS

Reviewer #1 (Remarks to the Author):

In this manuscript from the Dong lab, the authors explored the function of the BSL2, BSL3 and BSU1 phosphatases in stomatal development. The authors previously showed that BSL1, a member of the BSL family, is associated with the polarity complex on the cell membrane and activates the MAPKKK YODA in suppressing stomatal development. Here, using a combination of genetic, microscopic and biochemical approaches, they propose that the other three members of the family can function in the nucleus, and, in contrast to the role of BSL1, promote stomatal development by inhibiting nuclear MPK6.

The finding that the BSL phosphatases exhibit opposing functions in a cell compartment-dependent manner is novel and intriguing, and the authors presented a mechanism for the inhibitory effect in the nucleus. The work also helps dissect the role of individual BSLs in stomatal development. The experiments were generally well designed and executed. There are, however, a few points that I'd like the authors to address.

Response:

We thank the reviewer's appreciation of our work.

Major points: 1. Although the proposed inhibitory roles of BSL2 and 3 in stomatal development is mostly clear, the same can't be said for BSU1. The conclusion that BSU1 can suppress stomatal development (e.g. Line 78) is mainly based on the *bsl-q* mutant, which, as the authors stated, contains T-DNA insertions at BSU1 and BSL1 and an amiRNA construct(s) against both BSL2 and 3. Compared with T-DNA mutants, the amiRNA approach has a potential drawback in its off-target effect, i.e., in this case, towards unknown BSL-related genes, and causes artifacts. Indeed, based on the authors' data of the double and triple T-DNA mutants, mutations in BSU1 have never resulted in higher stomatal lineage index, but rather led to significant drops, which contradict with the *bsl-q* phenotype.

Response:

We thank the reviewer for sharing the insights about interpreting the mutant phenotypes. We are also thankful for the constructive suggestions that will greatly improve our data quality.

Here is what we understand about the quadruple mutant phenotypes. We do believe BSU1 contributes to the collective role of BSL1/BSL2/BSL3 at the PM to suppress stomatal formation, besides its positive role in the nucleus. This conclusion was based on the observations explained below.

- 1) Probably due to the mutagenesis feature (T-DNA combined with amiRNA), this quadruple mutant displays phenotypic severity at varying levels and shows a uniquely striking phenotype of **large patches of stomatal guard cells**, which often appear at the leaf edges (see below, blue arrows) but never observed in the triple mutant *bsl1;bsl2;bsl3*. On the other hand, when SLI (stomatal lineage index) is calculated, we have been conventionally using similarly central regions of the cotyledons to collect the numbers (red box below), where the large patches in the quadruple mutant were often missed for counting. This explains why ultimately the quantification of stomatal lineage index in the *quadruple* mutant was shown slightly higher than that of the triple mutant of

bsl1;bsl2;bsl3 (Fig. 1). We added more explanation in the main text to inform the mutant phenotype (see screen capture below).

stomatal production¹⁵ (Fig. 1b). To thoroughly determine the contribution of BSU1 in the BSL1/BSL2/BSL3-mediated negative regulation, we compared *bsl-q* with the null triple mutant *bsl1;bsl2;bsl3* and found that the quadruple mutant showed slightly increased stomatal production (quantified by stomatal lineage index, SLI, measured in the central region of adaxial side cotyledons in 5-day seedlings) (Fig. 1b). Moreover, the quadruple mutant was featured with severely clustered stomata often appearing as patches at the leaf edges (Supplementary Fig. 1a) that were not observed in the triple mutants. As the *bsl-q* mutant expresses *amiR-BSL2;3*

- 2) The other supporting evidence of BSU1, like BSL1/BSL2/BSL3, contributing to the negative regulation of stomatal formation at the PM came from the overexpression data of BSU1 vs. *myr-BSU1*. Indeed, the less nuclear partition of BSU1 (*myr-BSU1*) induced less severe stomatal overproduction *versus* the predominant nuclear BSU1 that induced severe stomatal overproduction (Fig. 2 and 3). The results suggested more BSU1 partition at the PM is associated with a negative regulation of BSU1 in stomatal formation.
- 3) To explain why *bsu1* mutation reduces stomatal production of the single or double mutations among *bsl1*, *bsl2*, or *bsl3* (Supplementary Fig. 2) but instead increased that of the triple mutant *bsl1;bsl2;bsl3*, we believe one should consider the signaling pathway of tiered YDA-MKK4/5-MPK3/6 cascade, so that the seemingly contradictory phenotype can be reconciled by the fact of YDA being an upstream activator of MPK3/6.

According to our model, the activation of upstream YDA is triggered by all four members (PM function), whereas the inhibition of MPK3/6 is controlled by BSL2/BSL3/BSU1 (nuclear function). In the quadruple mutant, the absence of four BSL members leads to the loss of YDA activity at the cell cortex, resulting in the loss of MPK3/6 activity in the nucleus. Therefore, in *bsl-quad*, regardless of the removal of BSL2/BSL3/BSU1-mediated inhibition on MPK3/6 in the nucleus, because of no YDA activating MPK3/6,

the overall phenotype in the quadruple mutant still shows the YDA loss of activity, thereby the enhanced overproduction of stomata in *bsl-q*. On the other hand, in the triple mutant *bsl2;bsl3;bsu1*, in which BSL1 is present and YDA is active, the removal of BSL2/BSL3/BSU1-mediated inhibition of MPK3/6 would lead to an enhanced MPK3/6 activity, resulting in suppressed stomatal production (as we see Fig. 1 and Supp. Fig. 2).

Thus, I suggest the authors to re-examine their data, and clarify the native roles of BSU1. Instead of relying solely on *bsl-q*, it would be ideal to assess the phenotype of T-DNA (or CRISPR)-based quadruple *bsl1/2/3 bsu1* mutants. Examining a single *bsl1* mutant with the same amiRNA against BSL2 and 3 (e.g. by crossing out the *bsu1* in *bsl-q*) and comparing it with *bsl-q* may also help address the effect of both the amiRNA and BSU1. Though, I also think the authors have sufficient genetic data that a careful re-assessment of them may suffice

Response:

We attempted to search for a null *bsl-quadruple* mutant from a crossed population (F2) of *bsl1;bsu1* X *bsl2;bsl3*, in which we identified all the mutation combinations in this study, but realized that the mission is impossible at this stage, because *BSU1* and *BSL2* are two closely linked genes (see below) and *bsl2 bsl3* double is sterile. Although we have germinated seeds to cross mutants *bsl1;bsl2;bsu1* with *bsl3* for the revision, we are not able to identify a true null quadruple mutant in a relatively short time window.

Indeed, the reviewer's suggestion of re-examination of the amiRNA lines is highly valuable. To rule out the possibility of off-target effect of amiRNA construct, as advised, we took strategies below.

- 1) First, we used qPCR to evaluate the transcript levels of *BSL1* and *BSU1* in the *amiRNA-BSL2/3* containing plants, i.e. *amiRNA-BSL2;3*, *bsl1 amiRNA-BSL2;3*, and *bsu1 bsl1 amiRNA-BSL2;3* (see data below). The results show that similar expression levels of *BSL1/BSU1* were observed in wild type vs. the silencing line *amiRNA-BSL2;3*, whereas their expression levels were almost under detection when the corresponding T-DNA insertional mutation was present. The results clearly demonstrated that no off-target effects of *amiR-BSL2;3* were observed on the homologous family members, *BSL1/BSU1*.

- 2) Furthermore, as advised, we fortunately have had *bsl1;amiRNA-BSL2;3* plants on hand. To evaluate the possible off-targeting of *amiRNA*, we compared stomatal phenotypes of (knock-down *amiRNA-BSL2;3* vs. null *bsl2;bsl3*) and (knockdown *bsl1;amiRNA-BSL2/3* vs. null *bsl1;bsl2;bsl3*). The results showed that the know-down mutants were comparable to or slightly weaker than the corresponding null mutants (see below). To consolidate the role of *bsu1* mutation, we compared stomatal production of *bsl1;amiRNA-BSL2;3* vs. *bsu1;bsl1;amiRNA-BSL2;3*. Indeed, the addition of *bsu1* enhanced the phenotype of *bsl1 amiRNA-BSL2;3* (see below). All these results support that *amiRNA-BSL2;3* is specifically knocking down the two target genes and *bsu1* enhances the stomatal overproduction phenotype caused by the combined mutations in *BSL1/BSL2/BSL3*.

- 3) Accordingly, we made major textual changes by adding a whole section to clarify the contribution of *BSU1* (see screen captures below).

stomatal production. While none of the single *bsl* mutants (T-DNA insertional lines) showed discernible stomatal and growth defects, our previous work also demonstrated that mutating more than two of the three BASL-interacting members, i.e. BSL1, BSL2, and BSL3, led to slightly increased stomatal production, whilst mutating all three led to moderately increased stomatal production¹⁵ (Fig. 1b). To thoroughly determine the contribution of BSU1 in the BSL1/BSL2/BSL3-mediated negative regulation, we compared *bsl-q* with the null triple mutant *bsl1;bsl2;bsl3* and found that the quadruple mutant showed slightly increased stomatal production (quantified by stomatal lineage index, SLI, measured in the central region of adaxial side cotyledons in 5-day seedlings) (Fig. 1b). Moreover, the quadruple mutant was featured with severely clustered stomata often appearing as patches at the leaf edges (Supplementary Fig. 1a) that were not observed in the triple mutants. As the *bsl-q* mutant expresses *amiR-BSL2;3* that silences the two close homologues (*BSL2* and *BSL3*), the possibility of off-targeting was ruled out by the detection of wild-type level of *BSL1* and *BSU1* transcripts in plants containing *amiR-BSL2;3* and by comparable phenotypes of *bsl1;bsl2;bsl3* and *bsl1;amiRNA-BSL2;3* (Supplementary Fig. 1b, c). Thus, our data indicated that BSU1 positively contributes to the BSL1/BSL2/BSL3-mediated inhibition of stomatal production in *Arabidopsis* (Supplementary Fig. 1a).

Further, I found the section describing the double/triple mutant phenotypes somewhat confusing, and improvements on its presentation would be needed.

Response:

We are thankful for the suggestion. After we added more details about characterization of the BSU1 contribution, the reading flows much better and helps the transition to the double and triple mutants. The edits were tracked below.

However, while we characterized all possible combinations of *bsl* double and triple mutations, surprisingly we found that differential combination of *bsl* mutations resulted in distinct phenotypes with regards to stomatal production (Fig. 1b, Supplementary Fig. 2). While *bsl1;bsl2;bsl3* produced more stomatal lineage cells¹⁵ (Fig. 1b), mutating the other three the members, i.e. *bsl2;bsl3;bsu1* led to an opposite phenotype - lowered stomatal production (Fig. 1b, Supplementary Fig. 2f, g, quantification based on stomatal lineage index). Combination of two mutations among these three genes (*BSL2*, *BSL3*, *BSU1*) also generated similarly lowered stomatal production in the mutants, e.g. *bsl2;bsu1* and *bsl3;bsu1*, (Fig. 1b). Thus, the results suggest that the combined function of *BSL2*, *BSL3* and *BSU1* confers an opposite regulation to promote stomatal production.

5

2. The *in vitro* assay that assessed the effect of BSL2 on MPK6 activity (Fig. 6a) used the phosphorylation status of MBP as a readout of MPK6 activity. However, the observed effect on MBP could be due to the BSL2 activity acting on the MBP protein itself. Thus, probing the effect on phosphorylated MPK6 (by anti-p42/44 or phos-tag gel) in the presence of BSL2 would be a superior readout and will better support the authors' claim that BSL2 acts on MPK6 directly. Further, the amount of the 5x BSL2 reaction seems off in the assay as judged by the anti-GST blot.

Response:

We thank the reviewer for pointing out this possibility. As suggested, we performed *in vitro* phosphorylation assays this time using MPK6 as substrate. The phosphorylation status of MPK6 was detected by anti-p42/44 antibody. Similar to what we found with the MBP substrate, we identified lowered MPK6 phosphorylation levels when more BSL2 is added in the system (data shown below). This time, the amount of BSL2 was better adjusted and shown by anti-GST blots.

3. The manuscript would be strengthened by having some initial investigations on whether the membrane/nuclear distribution of BSL2/3/BSU1 is dynamic or not. Given that BSL2/3 can interact with BASL, quantification of their distribution in stomatal lineage cells at different developmental stages, similar to what the authors have showed for BSL1 in their previous work, would be a reasonable start.

Response:

We thank the reviewer for sharing deep thoughts. As advised, we added the data of dynamic distribution of BSL2-mRFP when co-expressed with the polarity marker GFP-BASL. Results indeed showed the subcellular distribution of BSL2 can be dynamically changed at different stages in stomatal development (see data and textual additions below).

BSL phosphatases promote stomatal production in the nucleus ¶

To explicitly assay the functional contribution of the BSL proteins at the subcellular level, we first used BSL2, a family member functioning at both places, as an example to examine its nuclear/membrane partition in developing stomatal lineage cells. By co-expressing with the BASL polarity protein, we found BSL2-mRFP is more enriched in the nucleus when BASL is not polarized, whereas became more membrane associated when BASL is polarized (Supplementary Fig. xx). The results demonstrated the subcellular distribution so that function of BSL2 can be dynamically and spatially regulated by developmental regulators. Next, we

Other points:

- What is the transcript level of BSL2 and 3 in the *bsl-q* mutant compared with WT?

Response:

As requested, we checked the transcript level of BSL2 and 3 in the wild type and *bsl-q* mutant. The data is included part of Supplementary Fig. 1b.

- A few genetic materials and data (e.g. TMMp:BSL1-mRFP and localization of BSL2/3 and BUS1) were first appeared in the authors' last publication on BSL1. The authors should refer to the work when first presenting them to distinguish the novelties of the current study.

Response:

We thank the reviewer for careful reading of the manuscript. The references have now been included when the published materials were first appeared in this manuscript.

- The loading control (rubisco staining) in Fig. 6b is not clear. Since quantitative data were presented, it'd be better to represent loading by probing control proteins by Western blot.

Response:

As requested, the experiment was repeated to give improved results (see below). The Ponceau-S staining of Rubisco was replaced with Western Blotting using anti-MPK6 to show MPK6 protein levels *in vivo*.

Reviewer #2 (Remarks to the Author):

Stomata formation is regulated by a cascade of MAP kinases. The manuscript of Guo and Dong describes regulation of this cascade by four BSL phosphatases. Previously, this group showed that one of these phosphatases, BSL1, activates MAPKKK YODA at the plasma membrane. Here, they show that at the plasma membrane all four phosphatases dephosphorylate YODA which activates it. But in the nucleus, three out of the four phosphatases dephosphorylate a MAPK MPK6 which inactivates it. The authors clearly demonstrate that when subcellular localization of these phosphatases changes, their substrate changes too. This is an elegant paper that significantly contributes to our understanding of the regulatory mechanism of MAPK cascades, and it demonstrates how a phosphatase can function as both a positive and negative regulator of a signaling pathway. In addition, it provides a great example of how the nature of an enzyme's substrates can depend on the enzyme's subcellular localization. The data are comprehensive, of high quality and support the conclusions. The writing is clear and logical.

Response:

we appreciate the reviewer's positive feedback on the significance of our work.

Minor suggestions:

1. Figure 1B. Images (d) and (e) show different phenotypes. There is much more stomata clustering in (e) compared to (d). However, on the graph, the phenotype of these two mutants has no statistical difference. Are these images representative? Is SLI a good measurement of these phenotypes?

Response:

We thank the reviewer for careful reading of the manuscript.

As we elaborated above (in the Responses to Reviewer 1), the stomatal phenotypes of the quadruple mutants can vary in different regions of a leaf. We apologize for not providing the most representative image in the previous version to precisely match up the quantification data, which can be confusing for the audience. This has now been fixed with a more carefully selected image (see below) and additional textual explanation in the Results. Stomatal lineage index (SLI) has been widely used and is probably the most helpful and consistent measurement for evaluation of stomatal production because it measures both the early and late staged cells in the developmental process.

2. The use of asterisks in Fig 1B is confusing. A compact letter display would be better for visualization in this figure.

Response:

We thank the reviewer for the constructive suggestion. We have now added compact letters to represent the significance difference between groups shown in Figures 1b, 2c, 4k and Supplementary Fig. 2.

REVIEWERS' COMMENTS

Reviewer #1 (Remarks to the Author):

In the revised manuscript, the authors have performed additional genetic and biochemical analyses and provided detailed explanations to address my comments. I am satisfied with the revision.

Here are a few minor issues to note though:

1. For Fig. 1b (and other similar figures), the use of small capital letters (i.e., a, b, c..) now gets quite confusing as they represent sub-figures, genotypes and statistical differences. Perhaps the authors can use roman numeral (i.e., i, ii, iii...) for the genotypes instead.

2. Line 142-143, the sentence is not complete.

3. Fig. 4e and 4i appear to be the same sample with different views. Please us another image or delete one.

Reviewer #2 (Remarks to the Author):

Authors successfully addressed all raised concerns.

RESPONSE TO REVIEWERS' COMMENTS

Reviewer #1 (Remarks to the Author):

In the revised manuscript, the authors have performed additional genetic and biochemical analyses and provided detailed explanations to address my comments. I am satisfied with the revision.

We thank the reviewer for constructive suggestions that greatly improved our work.

Here are a few minor issues to note though:

1. For Fig. 1b (and other similar figures), the use of small capital letters (i.e., a, b, c..) now gets quite confusing as they represent sub-figures, genotypes and statistical differences. Perhaps the authors can use roman numeral (i.e., i, ii, iii...) for the genotypes instead.

As suggested, we have replaced with roman numerals for the genotypes.

2. Line 142-143, the sentence is not complete.

The sentence has been corrected.

“By co-expression with the BASL polarity protein¹⁵, we found that BSL2-mRFP is more enriched in the nucleus when BASL is not polarized, but BSL2-mRFP became more membrane associated when BASL is polarized (Supplementary Fig. 4).”

3. Fig. 4e and 4i appear to be the same sample with different views. Please us another image or delete one.

As suggested, we replaced the image used for Figure 4i.

Reviewer #2 (Remarks to the Author):

Authors successfully addressed all raised concerns.